# Biodegradation of Aqueous Superabsorbents: Kinetic Assessment Using Biological Oxygen Demand Analysis

Andrey V. Smagin [1,2,*], Nadezhda B. Sadovnikova [1] and Viktor I. Budnikov [2]

1   Soil Science Department and Eurasian Center for Food Security, Lomonosov Moscow State University, GSP-1, Leninskie Gory, 119991 Moscow, Russia
2   Institute of Forest Science, Russian Academy of Sciences (ILAN), 21, Sovetskaya, Uspenskoe, 143030 Moscow, Russia
*   Correspondence: smagin@list.ru; Tel.: +7-(495)916-913-79-48

**Abstract:** Biodegradation is an important environmental factor controlling the effectiveness of aqueous superabsorbents for soil conditioning. The purpose of the study is to quantify this process using biological oxygen demand (BOD) analysis of composite superabsorbents with an acrylic polymer matrix, amphiphilic fillers (humates, peat) and silver ions as an inhibitor of biological activity. A simple kinetic model of BOD is proposed for standardization of the analysis and calculation of polymer half-life after their long-term (60–120 days) incubation in the VELP BOD analyzer (Italy) with automatic control. The half-life of pure hydrogels pre-swollen in distilled water (1:100) at 30 °C varied from $0.8 \pm 0.2$ to $2.4 \pm 1.6$ years. The addition of water extract from compost sharply enhances the biodegradation, lowering the half-life up to 40–60 days. Doses of 0.1–1% silver in a polymer matrix or 10–100 ppm in swollen hydrogels increase their half-life by 5–20 times. The discussion part questions the traditional division of aqua superabsorbents into "biodegradable" and "non-biodegradable", and also analyzes the main advantages and disadvantages of the new methodology for their BOD analysis. The results may be of interest to a wide range of specialists from chemical technologists and biochemists to environmental engineers.

**Keywords:** biodegradation; composite materials; gel-forming superabsorbents; incubation; biological oxygen demand; BOD respirometric sensors; process modeling

## 1. Introduction

Biodegradation determines the lifetime and functioning of natural and synthetic polymers in the environment. Many factors control biodegradation—from the chemical composition and structure of materials, to environmental factors of temperature, humidity, pH, redox potential, dispersity, the presence of destructor organisms and their enzymes, and the depth of localization of polymers in the soil [1–4]. As a result, even chemically stable organic materials can decompose rather quickly, occurring in biologically and chemically active environments, for example, soils, composting devices, air and methane tanks of municipal water treatment [1,2,5]. The assessment of biodegradability is extremely relevant for artificially created polymers and composites, since it underlies the scientific prediction of their service life and usefulness in the environment. Despite the current standards of biodegradability in Europe and the USA (EN 13432, ASTM 6400 [3]), only a few synthetic materials are subject to mandatory certification with a quantitative assessment of the biodegradation rate in the form of their half-life in natural and anthropogenic landscapes. As a rule, only synthetic pesticides pass the mandatory test of resistance to biodegradation in laboratory and field conditions [1,6]. For soil conditioners, including polymeric composite hydrogels (aqueous superabsorbents), similar studies are still quite rare [2,7,8]. Therefore, the available information here is extremely contradictory, and the traditional division of these materials into "biodegradable" (usually biopolymer hydrogels based on polysaccharides and their composites) and "non-biodegradable" (usually

polyacrylamide, acrylic acrylonitrile, and other synthetic hydrogels) is being increasingly questioned [2,7,9–12]. For the object of our research, acrylic composite superabsorbents, a number of publications report extremely high resistance to biodegradation [12,13], while other sources refute these facts, reporting the rapid biodegradation of these materials in the environment [2,7,8].

Such an uncertain situation can be explained, in our opinion, by the lack of unified adequate methods and models for quantifying the biodegradability of organic materials. Traditional quantification usually uses polymer weight dynamics or gasometric analysis ($CO_2$ release, $O_2$ consumption) during biodegradation in laboratory or, less often, field conditions [14–18]. The criteria for biodegradation in the vast majority of accepted international tests (OCDE, ISO) are the percentage of elimination of organic carbon during the experiment [14,15]. Less often, kinetic estimation of organic carbon dynamics is used mainly with empirical first-order or quasi-first-order kinetic models, the Monod model, and the Ferhulst–Perl (logistic) model [14,17,18]. Basically, kinetic estimation operates on the rate of weight loss of organic matter over time or synchronous release of carbon dioxide. Attempts to use biological oxygen demand (BOD) analysis to assess the biodegradation of organic materials are less common, and most often BOD is used only as an indicator of water quality or for a rough evaluation of the content of biodegradable organic substances in the aquatic environment [5,19]. Here, again, despite the possibility of automated monitoring of the BOD curves in real time using modern BOD sensors (for example, WTW OxiTop IS BOD sensor [20] or VELP BOD sensor [21]), the main resulting indicators are only one-time values of the BOD (for example, $BOD_5$ or $BOD_{10}$ for 5 or 10 days of the experiment, respectively) [19]. The absence of a generally accepted kinetic theory of the BOD experiment complicates the interpretation of the results with the transformation of the obtained BOD curves into the biodegradation curves of organic materials for an accurate assessment of their half-life.

In this regard, the purpose of our study was to quantify the biodegradability of composite aqueous superabsorbents for soil conditioning using automatic manometric BOD analyzers. The main tasks included:

○ development of the kinetic theory of BOD analysis for assessing the biodegradability of gel-forming organic materials;

○ obtaining kinetic BOD curves for the studied hydrogels in long-term (60–120 days) incubation experiments based on the VELP BOD analytical equipment;

○ mathematical processing of BOD curves for calculating the half-life indicators of the studied aqueous superabsorbents;

○ comparative analysis of the obtained results and controlling factors (composition, biocidal additives, incubation conditions) of biodegradability for gel-forming soil conditioners.

The new methodology and automatic VELP BOD analytical equipment were used for experimental evaluation of the biodegradation process in composite gel-forming soil conditioners, apparently for the first time. This study is significant and valuable from the point of view of the technological chemistry of composite gel-forming superabsorbents, identifying the dependence of their biodegradability not only on the chemical composition, but primarily on the conditions of the incubation experiment. To show this, along with the usual incubation of pure hydrogels swollen in distilled water, we used the addition of a compost extract with microorganisms to the liquid phase of hydrogels in order to stimulate biodegradation and bring the laboratory experiment closer to real soil conditions. To effectively reduce the biodegradability of composite aqueous superabsorbents, we propose, apparently for the first time, the incorporation of silver biocides into their polymer matrix. The results of the study, contrary to the traditional opinion, show the possibility of rapid biodegradation for acrylic superabsorbents, which is important both for specialists in the field of their chemical synthesis and for practitioners for use in soil engineering and precision irrigation.

## 2. Materials and Methods

### 2.1. Tested Composite Gel-Forming Materials

Studied composite materials were synthesized at the Ural Chemical factory (Russian Federation, Perm) under the "Aquapastus" trademark using our patented technology [22] (see also Section 6 at the end of the article). These polymeric material products included various compositions of acrylic copolymers based on acrylamide and acrylic acid salts, filled by wastes of biocatalytic production of acrylamide, salts of humic acids, and dispersed peat. Some variants also included the addition of ionic silver to the polymer matrix to protect the composites from biodegradation. Methylene-bis-acrylamide was used as a crosslinking agent. The water absorption of new products by swelling in distilled water varied from 340 to 500 kg/kg for granules with sizes near 1 mm. The "Aquapastus"-11 (the A11) hydrogel is the base co-polymer of acrylamide and ammonium acrylate filled (28%) by solid wastes of a biocatalytic production of acrylamide as a mixture of microbial cells, cell agglomerates and filtroperlit. The formulation "Aquapastus"-11Hm (the A11-Hm) includes, in addition to biocatalytic wastes (12%), humates of potassium and sodium, amounting to 8% of dry matter. Their modifications included 0.1–1% additives of silver ions in the form of nitrate (the A11Ag or A11HmAg gels) as inhibitor of polymer's biodegradation. The last two composites, A22 and A22Ag, along with the co-polymer of acrylamide and sodium acrylate, contained a finely dispersed peat as a filler (23.5%) and 0.1–1% additives of ionic silver. In laboratory experiments, the new products described above were compared with the well-known "Aquasorb" brand manufactured by SNF-group [23]. It is a superabsorbing anionic polymer in the form of crosslinked copolymers of acrylamide and potassium acrylate, characterized by a maximum degree of swelling of at least 500 kg $H_2O$/kg for a granule size of 0.2–0.8 mm (Aquasorb 3005KM). Detailed descriptions of the synthesis, composition, and results of preliminary laboratory testing of the Aquapastus composites for soil conditioning are presented in our previous articles [8,22] and patents (see Section 6 at the end of the article).

### 2.2. BOD Analysis Method and VELP Equipment

To quantify the biodegradability of composite gel-forming materials, we chose BOD analysis based on VELP Scientifica equipment, namely the VELP RESPIROMETRIC Sensor System 6 for Soil Analysis (VELP Scientifica, Usmat, Italy) [21]. The advantage of this assay is the ability to obtain a continuous curve of biological oxygen uptake in a long incubation experiment. The analysis principle consists in continuous monitoring of the gas pressure inside a hermetically sealed vial using a sensitive automatic VELP sensor, pre-programmed for the required frequency of pressure measurements. During aerobic biodegradation of organic matter in a closed measuring system, oxygen is absorbed and carbon dioxide is released from decaying material. The released $CO_2$ is captured by alkali (KOH), so the gas pressure in a closed system is continuously decreasing, in proportion to the biological absorption of oxygen. The manometric gas pressure reduction curve recorded by the VELP sensor is easily transformed into a BOD curve, which can be used to calculate the kinetic constants of biodegradation and half-life indicators for organic materials according to our approach (see the theory of BOD analysis in Section 3.1).

The manometric BOD analysis based on VELP Scientifica equipment is performed completely automatically. Using the wireless control device DataBox™ and RESPIROSOFT™ 1.0.0 Software [21], the VELP respirometric sensor transmits data directly to the PC enabling real-time monitoring of the analysis curve. This is very convenient, especially for long-term (several months) incubation experiments, which are required in the case of biodegradation-resistant synthetic polymers. Unlike the well-known analogues that measure BOD mainly in liquid media (such as OxiTop sensors (Germany, WTW, Weilheim in Oberbayern, [20])), VELP equipment includes a line of analyzers for polyphase heterogeneous media, for example, VELP RESPIROMETRIC Sensor System 6 for Soil Analysis used in our study. The conical shape of VELP incubators (vials) with a wide corrugated bottom improves gas exchange, contributing to the rapid absorption of oxygen from the air space of the vial by a

thin layer of incubated material evenly distributed over a wide bottom. In order to calculate the optimal amount of such material in the case of polymeric hydrogels, we propose to standardize the BOD analysis on the basis of appropriate kinetic models (see Section 3.1 for details). The main idea of standardization was to use a fixed mass of the incubated hydrogel, so that the number of carbon moles in this mass was equal to the number of oxygen moles in the air volume of the BOD analyzer.

For incubation experiments, this calculated amount of dry gel was added to the aqueous liquid phase in a ratio of 1:100 to obtain swollen gel structures. We used two types of liquid phase for swelling—pure distilled water and distilled water with an aqueous extract from compost consisting of rotting vegetables and fruits (potatoes, onions, apples, grapes, oranges). Unlike pure distilled water, compost extract should contain specific microorganism-destructors and, possibly, their exoenzymes. The addition of such "compost yeast" should, in our opinion, bring the laboratory experiment closer to real soil conditions and stimulate the biodegradation of the studied synthetic polymers. The gel structures were placed in 1 L incubation vials of the VELP RESPIROMETRIC Sensor System 6 for Soil Analysis [21]. Simultaneously, a $CO_2$ absorber in the form of granulated chemically pure potassium hydroxide (Russia, EKOS-1, Moscow, [24]) was placed in a rubber holder in the upper part of the vial. After that, the vial was hermetically sealed with a lid containing the VELP BOD Sensor for automated BOD manometric analysis. Closed vials were installed in a BINDER ED023-230V thermostat (BINDER GmbH, Tuttlingen, Germany) for a long-term experiment on the incubation of gel structures at a constant temperature of 30 °C, optimal for aerobic biodegradation.

Preliminary adaptation of the BOD analysis method based on the VELP RESPIROMET-RIC Sensor System 6 for Soils showed that obtaining representative BOD curves in the case of composite gel-forming soil conditioners requires a long incubation time of 60–120 days. In this case, the frequency of measurements can be reduced to 1 time per day or per 2 days, in order, on the one hand, to guarantee the representativeness of the measurements, and, on the other hand, not to exceed the memory capacity of the BOD sensor. For long-term analyses, only direct pressure measurement is available in the VELP RESPIROMETRIC Sensor System 6 for Soil Analysis; however, these data are subsequently easily converted into a BOD kinetic curve, which is necessary for evaluating the biodegradability of materials.

Necessary mathematical and statistical processing of the results, including data approximation by nonlinear thermodynamic models, was carried out using MS Excel, Microsoft Office 2016 (Microsoft, Redmond, DC, USA) and S-Plot 11 (Systat Software GmbH, Erkrath, Germany) computer software.

## 3. Results

### 3.1. Kinetic Theory of BOD Analysis for Assessing the Biodegradability of Gel-Forming Organic Materials

Conventional BOD analysis uses the incubation of organic matter in closed vials with automatic control of oxygen uptake dynamics. Regardless of the oxygen uptake rate, the maximum BOD ($U_m$, [g/m$^3$ = mg/L]) in a closed vial is obviously equivalent to the atmospheric oxygen concentration ($C_0$, [g/m$^3$ = mg/L]). This indicator depends on barometric pressure ($P$, [Pa]) and air temperature ($T$, [K]) according to the Mendeleev–Clapeyron law:

$$C_0 = \frac{20.9 \cdot P \cdot M}{100 \cdot R \cdot T}, \tag{1}$$

where 20.9 [%] is the volume ratio of oxygen in the atmosphere; $M$ = 32 g/mol is the molar mass of oxygen; $R$ = 8.314 J/(mol K) is the universal gas constant. For the usual temperature range of 293–303 K and barometric pressure of 96–104 kPa, the $C_0$ index varies from 255 to 285 g/m$^3$ with an average value of 270 g/m$^3$.

The specific parameter $U_m$ can vary greatly depending on the volume ($V_s$, [g/m$^3$]) of the studied substance (solution, gel, powder):

$$U_m = \frac{C_0 \cdot (V_0 - V_s)}{V_s}, \tag{2}$$

where $V_0$, [g/m$^3$] is the volume of the BOD gas-analyzer vial. However, the maximum mass ($m_0$, [g]) of oxygen available for biological absorption during incubation in a closed vial obviously cannot be greater than $m_0 = C_0 \cdot V_0$.

Under conditions of oxygen limitation, the BOD dynamics can be described by the following kinetic model:

$$\frac{dU}{dt} = -b(U_m - U), \tag{3}$$

where $U$ [g/m$^3$ = mg/L] is the current BOD; $b$ [day$^{-1}$] is the kinetic constant. If oxygen is over ($U = U_m$), the rate of its biological absorption is zero ($dU/dt = 0$).

The solution of differential Equation (3) provides a simple relaxation model for describing the BOD dynamics in time:

$$U_{(t)} = U_m(1 - exp(-b \cdot t)). \tag{4}$$

In aerobic biodegradation of organic matter, one mole of oxygen binds one mole of carbon, producing one mole of carbon dioxide ($CO_2$). In this case, Equation (4) is transformed directly into the material decomposition equation:

$$X_{(t)} = X_0 - \frac{12}{32} \frac{U_m}{\rho_s} (1 - exp(-b \cdot t)), \tag{5}$$

where $X_0$, $X_{(t)}$, [kg/kg] are the concentrations of organic carbon in the material at the beginning of the experiment ($t = 0$) and at subsequent stages; 12/32 is the ratio of molar masses of carbon and oxygen, $\rho_s$, [kg/m$^3$] is the density of the material.

Equations (1), (2) and (5) allow us to standardize the aerobic BOD analysis so that the amount of carbon in organic matter is always equivalent to the maximum amount of oxygen in the closed vial of the BOD analyzer. The standardization condition is

$$X_0 = \frac{12}{32} \frac{U_m}{\rho_s}. \tag{6}$$

In this case, Equation (5) is transformed into the standard exponential model of biodegradation [2,14]:

$$X_{(t)} = X_0 \cdot exp(-b \cdot t). \tag{7}$$

The half-life ($T_{0.5}$, [day]) of a material can be calculated directly from this model using the kinetic BOD constant ($b$):

$$T_{0.5} = \frac{\ln(2)}{b}. \tag{8}$$

Taking into account Equation (2) and the condition $V_0 \gg V_s$, it is easy to transform (6) into the following formula for determining the standard amount of material in BOD analysis:

$$m_s \approx \frac{12}{32} \frac{C_0 \cdot V_0}{X_0}, \tag{9}$$

where $m_s$, [g] = $\rho_s \cdot V_s$ is the standard mass of the material, equivalent to the mass of atmospheric oxygen (oxidant) in the closed vial of the BOD analyzer. For example, for a liter vial ($V_0 = 1$ L) with an average oxygen content in the laboratory atmosphere $C_0 = 267$ g/m$^3$ = 267 mg/L, the standard mass of acrylic composites with a carbon content of 40–50% ($X_0 = 0.4$–0.5 g/g) should be equal to 200–250 mg according to Formula (9).

Since organic materials, especially superabsorbent polymeric hydrogels, are highly hygroscopic substances [25], it is necessary to take into account the content of hygroscopic water ($W_h$, [%]) when calculating the standard mass for BOD analysis. In this case, Formula (9) is transformed into the following equation:

$$m_s \approx \frac{12}{32} \frac{C_0 \cdot V_0}{X_0} \cdot \frac{(100 + W_h)}{100}. \tag{10}$$

The use of standard masses of organic matter in the aerobic BOD analysis makes it possible to avoid the inhibition of biodegradation by oxygen deficiency, as well as to evaluate the half-life of the material directly from the kinetic BOD curves after their approximation by Equation (4).

Since the long-term BOD analysis uses direct monitoring of gas pressure in a closed vial, it is possible to simplify Model (4) for BOD dynamics by replacing the specific BOD with the content of absorbed oxygen in the vial ($C_{(t)}$, [g/m$^3$]), measured by the decrease in pressure ($\Delta P(t)$, [Pa]):

$$C_{(t)} = C_0(1 - exp(-b \cdot t)) = \frac{20.9 \cdot M}{100 \cdot R \cdot T} \Delta P_{(t)}. \tag{11}$$

In this case, the accuracy of the manometric BOD analyzer (0.355 kPa) provides an estimate of the oxygen concentration $C_{(t)}$ in increments of 1 g/m$^3$ in the range from 1 to 269–278 g/m$^3$ (mg/L) for an incubation temperature of 293–303 K and a standard atmospheric pressure of 101.3 kPa.

For materials resistant to biodegradation with an almost linear trend $C(t)$ at the maximum incubation interval of 0–120 (180) days, it is convenient to expand Function (11) into a Maclaurin series:

$$C_{(t)} = C_0(1 - exp(-b \cdot t)) \approx C_0 \cdot b \cdot t. \tag{12}$$

In this case, the approximation of experimental data by a linear function $C(t) = a \cdot t$ allows us to easily estimate the kinetic constant of biodegradation:

$$b = \frac{a}{C_0}. \tag{13}$$

Parameter $C_0$ is calculated by Formula (1). The constant $b$ obtained in this simplified way is used further to evaluate the half-life of the material, according to Equation (8).

As an example, Figure 1 shows a fragment of automatic monitoring of pressure dynamics ($P(t)$) in the process of oxygen absorption by the Aquasorb hydrogel with compost extract addition, as well as the corresponding dynamics of the absorbed oxygen content ($C(t)$) calculated by Formula (11). This fragment corresponds to the initial (linear) part of the BOD curve. Therefore, a simplified linear Model (12) for BOD dynamics can be used to approximate it. The linear trend parameter or straight line slope ($a$) for this example was $a = 0.1834$ g/(m$^3$ h). The initial oxygen content in the vial ($C_0$) at an atmospheric pressure of 100.4 kPa (1004 hPa) and an experimental temperature of 30 °C (303 K) was 267 g/m$^3$, according to Formula (1). In this case, the calculation of the kinetic constant of biodegradation, according to Formula (13), provides $b = 0.1834/267 \times 24 \times 365 \approx 6$ yr$^{-1}$. Here, we assumed that biodegradation lasts at the same rate for a whole year (365 days). This assumption obviously provides the maximum estimate of the intensity of biodegradation, since in real conditions it can be restrained by cold temperatures or even interrupted in the winter season. Having a constant estimate of the biodegradation, it is easy to calculate the half-life value of the studied polymer material using Formula (8). This indicator did not exceed $T_{0.5} = 0.12$ years or less than 1.5 months. That is, under conditions of an optimum temperature of 30 °C and the presence of soil microorganisms-destructors, the Aquasorb hydrogel loses a half of its mass in less than 1.5 months due to biodegradation.

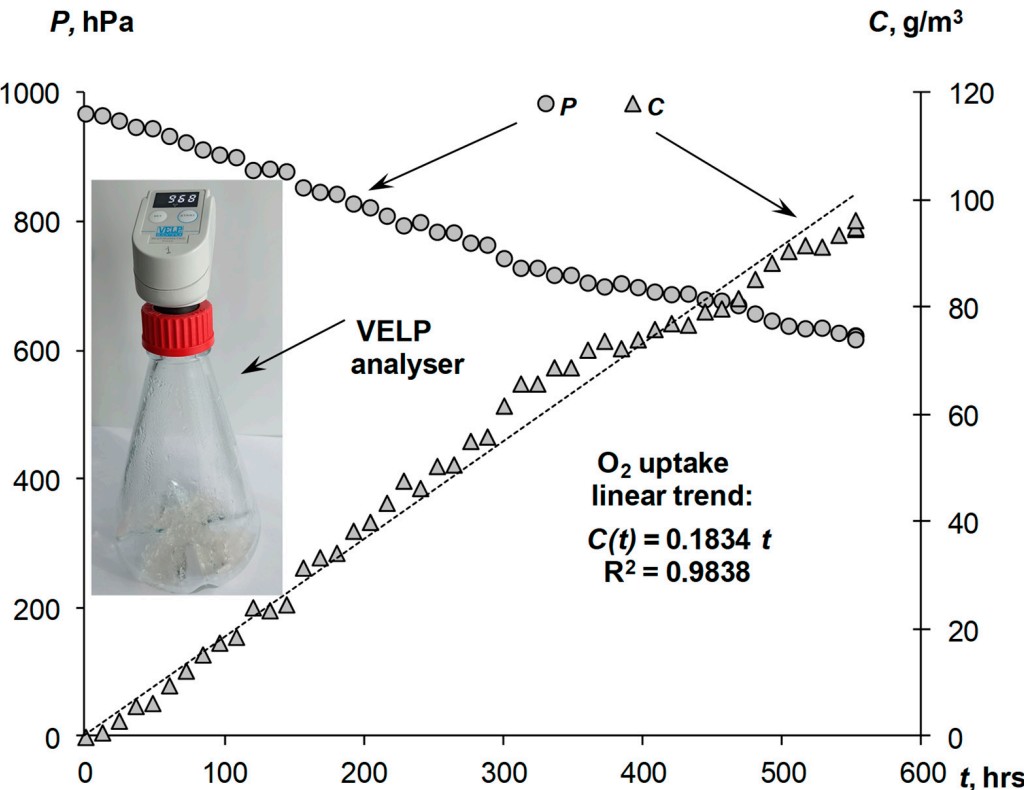

**Figure 1.** Experimental trend of pressure dynamics $P(t)$ transformed according to Formula (11) into an oxygen uptake curve ($C(t)$) (main figure); the VELP Respirometer with the Aquasorb gel (inset photo).

*3.2. Experimental Results*

3.2.1. Respirometric BOD Curves

Figure 2 shows the results of an instrumental assessment of BOD for gel-forming soil conditioners with an acrylic polymer matrix. The most significant differences in the respiration curves are observed between gels swollen in distilled water and in a solution containing compost extract. Standard BOD value (decrease in oxygen concentration) in pure gels did not exceed 50 g/m$^3$, while the addition of compost extract increased BOD to 150–200 g/m$^3$ and more, that is, 3–4 times compared to pure samples with the same standard mass. Against this background, the influence of the composition of superabsorbents on BOD was in most cases less significant. The maximum BOD values are observed for the A11 material with amphiphilic additives from the wastes of the biocatalytic production of polyacrylamide. They reach 70 and 250 g/m$^3$ for samples swollen in distilled water and compost solution, respectively. The minimum BOD values were detected in the composite superabsorbent A22 with an amphiphilic filler in the form of dispersed peat. BOD in pure A22 gel did not exceed 22 g/m$^3$, and in the variant with compost extract it was 205 g/m$^3$. The maximum BOD values for other types of gels occupied intermediate positions. Thus, the largest differences between BOD for superabsorbents with different chemical composition did not exceed 70/22 = 3.2 times for pure aqueous solutions and 250/205 = 1.2 times for the liquid phase with compost extract.

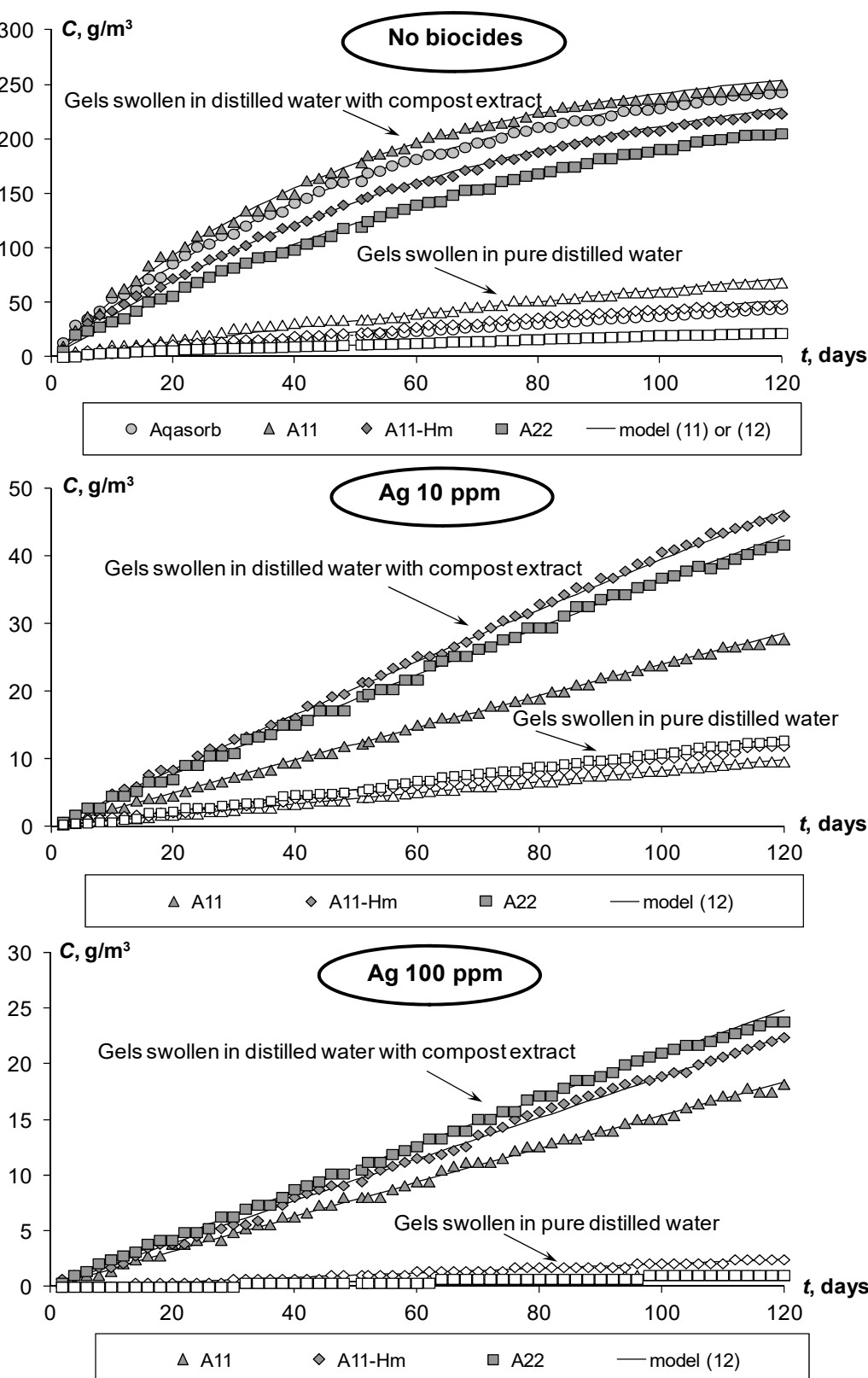

**Figure 2.** Respirometric oxygen uptake curves (*C*(*t*)) for various types of composite hydrogels and different incubation conditions (with and without compost extract). For the names of the gels, see "Objects and methods". Ag 10 ppm, Ag 100 ppm are the concentrations of silver ions in gel structures at a swelling degree of 1:100.

The incorporation of ionic silver into the acrylic polymer matrix effectively reduces the BOD of the composite gel-forming materials according to the dose of biocide (Figure 2, middle and lower parts). A relatively small dose of 0.1% (10 ppm Ag in a 1:100 swollen gel structure) causes a decrease in maximum BOD values at the end of incubation up to 10–13 $g/m^3$ for pure gels and to 28–46 $g/m^3$ for gels swollen in compost extract. The drop in BOD reaches 7–9 times that of the original gels without biocide (Figure 2). This effect overlaps the differences between the variants with distilled water and compost extract, which in this case do not exceed 3–3.5 times.

The use of a higher dose of 1% silver (100 Ag in a 1:100 swollen gel structure) enhances BOD inhibition (Figure 2 lower part). For variants with compost extract, the resulting BOD at the end of incubation did not exceed 18–24 $g/m^3$. This meant a reduction in BOD by 10–15 times under the action of the biocide in comparison with similar variants of composites without silver. For gels swelling in distilled water, the BOD under the influence of the biocide decreased up to the sensitivity limit of the BOD respirometric method (1–2 $g/m^3$) or more than 20–30 times.

### 3.2.2. BOD Analysis and Evaluation of the Biodegradability of Composites

The shapes of the experimental respirometric curves $C_{(t)}$ were visually different (Figure 2). In case of high values of the standard BOD (200–250 $g/m^3$ or more), they had a convex shape, as follows from the theory of BOD analysis (Model (4)). However, for materials with small BOD (20–50 $g/m^3$ and below), respirometric curves corresponded to a linear trend to a greater extent. Therefore, we applied in their mathematical processing both models of BOD analysis, presented in the theoretical part by Equations (11) and (12). The parameters of the approximation of experimental data by these models and the corresponding statistics are presented in Table 1. Their analysis shows a rather high adequacy of both models to the experimental data. Determination coefficients ($R^2$) in most cases reached values of 0.996–0.999 with small standard error of estimate (0.2–3.2 $g/m^3$). However, for linear experimental trends with low BOD, the two-parameter model (11) often provided statistically unreliable approximation parameters with unacceptable confidence intervals and $p$-value (0.1427–0.9886). In these cases, the calculations of the kinetic constant ($b$) for the assessment of material half-life were carried out using a simplified linear version of the BOD Model (12). Conversely, for curves with high final BOD values, the linear Model (12) was inadequate, according to low $R^2$ values (0.716–0.851) and high standard errors (s = 25–32 $g/m^3$). Therefore, in such cases, it was necessary to use the basic Model (4) for mathematical data processing and obtaining half-life estimates.

**Table 1.** Approximation parameters of Models (11) and (12) for standardized BOD curves.

| Liquid Phase | Nonlinear Model (11) | | | | | Linear Model (12) | | | |
|---|---|---|---|---|---|---|---|---|---|
| | $R^2$ | s | $C_0$ | $10^3 \cdot b$ | $p$-Value | $R^2$ | s | $a = C_0 \cdot b$ | $p$-Value |
| Aquasorb | | | | | | | | | |
| Com.ext. * | 0.998 | 2.2 | $278 \pm 2$ | $17.8 \pm 0.3$ | 0.0001 | 0.783 | 31.7 | $2.494 \pm 0.585$ | 0.0001 |
| DW ** | 0.999 | 0.4 | $278 \pm 33$ | $1.5 \pm 0.2$ | 0.0001 | 0.996 | 0.8 | $0.385 \pm 0.015$ | 0.0001 |
| A11 | | | | | | | | | |
| Com.ext. | 0.998 | 3.2 | $278 \pm 2$ | $20.3 \pm 0.3$ | 0.0001 | 0.716 | 37.6 | $2.631 \pm 0.693$ | 0.0001 |
| DW | 0.981 | 2.8 | $278 \pm 70$ | $2.5 \pm 0.7$ | 0.0008 | 0.962 | 3.9 | $0.629 \pm 0.071$ | 0.0001 |
| A11-Ag 10 ppm | | | | | | | | | |
| Com.ext. | 0.999 | 0.3 | $278 \pm 56$ | $0.9 \pm 0.2$ | 0.0001 | 0.998 | 0.4 | $0.239 \pm 0.007$ | 0.0001 |
| DW | 0.998 | 0.2 | $278 \pm 206$ | $0.3 \pm 0.2$ | 0.0008 | 0.998 | 0.1 | $0.083 \pm 0.002$ | 0.0001 |

**Table 1.** *Cont.*

| Liquid Phase | Nonlinear Model (11) | | | | | Linear Model (12) | | | |
|---|---|---|---|---|---|---|---|---|---|
| | $R^2$ | s | $C_0$ | $10^3 \cdot b$ | *p*-Value | $R^2$ | s | $a = C_0 \cdot b$ | *p*-Value |
| A11-Ag 100 ppm | | | | | | | | | |
| Com.ext. | 0.997 | 0.3 | $278 \pm 140$ | $0.6 \pm 0.3$ | 0.0583 | 0.996 | 0.3 | $0.156 \pm 0.006$ | 0.0001 |
| DW | 0.913 | 0.1 | $289 \pm 686$ | $0.03 \pm 2.0$ | 0.9874 | 0.913 | 0.1 | $0.009 \pm 0.002$ | 0.0001 |
| A11-Hm | | | | | | | | | |
| Com.ext. | 0.997 | 2.9 | $278 \pm 3$ | $14.3 \pm 0.3$ | 0.0001 | 0.851 | 24.7 | $2.245 \pm 0.454$ | 0.0001 |
| DW | 0.987 | 1.6 | $278 \pm 90$ | $1.7 \pm 0.6$ | 0.0061 | 0.943 | 2.1 | $0.441 \pm 0.038$ | 0.0001 |
| A11-Hm-Ag 10 ppm | | | | | | | | | |
| Com.ext. | 0.999 | 0.5 | $278 \pm 31$ | $1.5 \pm 0.2$ | 0.0001 | 0.997 | 0.7 | $0.402 \pm 0.014$ | 0.0001 |
| DW | 0.997 | 0.2 | $278 \pm 206$ | $0.4 \pm 0.3$ | 0.1891 | 0.997 | 0.2 | $0.100 \pm 0.003$ | 0.0001 |
| A11-Hm-Ag 100 ppm | | | | | | | | | |
| Com.ext. | 0.998 | 0.3 | $278 \pm 96$ | $0.7 \pm 0.3$ | 0.0071 | 0.998 | 0.3 | $0.191 \pm 0.006$ | 0.0001 |
| DW | 0.989 | 0.1 | $289 \pm 353$ | $0.1 \pm 0.9$ | 0.9353 | 0.979 | 0.1 | $0.020 \pm 0.002$ | 0.0001 |
| A22 | | | | | | | | | |
| Com.ext. | 0.998 | 2.3 | $278 \pm 3$ | $11.6 \pm 0.2$ | 0.0001 | 0.917 | 17.4 | $2.011 \pm 0.321$ | 0.0001 |
| DW | 0.952 | 1.4 | $278 \pm 375$ | $0.8 \pm 1.1$ | 0.4773 | 0.943 | 1.5 | $0.205 \pm 0.027$ | 0.0001 |
| A22-Ag 10 ppm | | | | | | | | | |
| Com.ext. | 0.998 | 0.5 | $278 \pm 44$ | $1.4 \pm 0.2$ | 0.0001 | 0.996 | 0.8 | $0.365 \pm 0.015$ | 0.0001 |
| DW | 0.998 | 0.2 | $278 \pm 184$ | $0.4 \pm 0.3$ | 0.1427 | 0.997 | 0.2 | $0.109 \pm 0.004$ | 0.0001 |
| A22-Ag 100 ppm | | | | | | | | | |
| Com.ext. | 0.998 | 0.53 | $278 \pm 79$ | $0.8 \pm 0.2$ | 0.0013 | 0.997 | 0.4 | $0.208 \pm 0.007$ | 0.0001 |
| DW | 0.905 | 0.1 | $289 \pm 683$ | $0.03 \pm 2.1$ | 0.9886 | 0.905 | 0.1 | $0.009 \pm 0.002$ | 0.0001 |

* the liquid phase is distilled water and compost extract (Com.ext.); ** the liquid phase is pure distilled water (DW).

### 3.2.3. Half-Life of Acrylic Composite Superabsorbents

The calculated half-life rates for various gel-forming soil conditioners and variants of the intermicellar solution for their swelling are shown in Figure 3. They vary widely from 35–65 days to 10–30 years or more, depending on the chemical composition of the materials and incubation conditions (composition of the liquid phase for swelling). The common opinion in polymer chemistry about the high stability of acrylic polymer hydrogels belonging to the class of "non-biodegradable" [3,7–12] is confirmed when using distilled water as a liquid phase medium for the preparation of gel structures. In this case, the half-life of pure gels (without biocides) varies from $0.81 \pm 0.06$ years (A11 material with hydrophilic biocatalytic fillers) to $2.50 \pm 0.06$ years (A22 composite with amphiphilic peat filler). For the Aquasorb hydrogel, the half-life is $1.33 \pm 0.05$ years, which, in terms of 95% degradation according to the standard exponential model (7), offers the life of the polymer in the soil of about 5.5–6 years. However, all of these estimates are based on clean, near-sterile laboratory conditions.

The addition of a small amount of an aqueous extract from compost to the swelling solution sharply (by 8–15 times) enhances the biodegradation of the studied polymeric materials. For the most stable A22 composite, half-life is reduced by 15 times up to $0.17 \pm 0.06$ years, i.e., 50–70 days. The half-life of other polymeric materials is even less and varies from $34 \pm 11$ days for the most biodegradable A11 gel to $48 \pm 11$ days for the A11-Hm composite with an amphiphilic humate filler. The Aquasorb gel in this case has a half-life of $39 \pm 10$ days, that is, 12 times less than the initial indicator for the gel structure in distilled water. All these results, in our opinion, cast doubt on the idea of high stability of

acrylic polymer hydrogels in the real soil environment and initiate the challenge of creating more stable composites for soil conditioning.

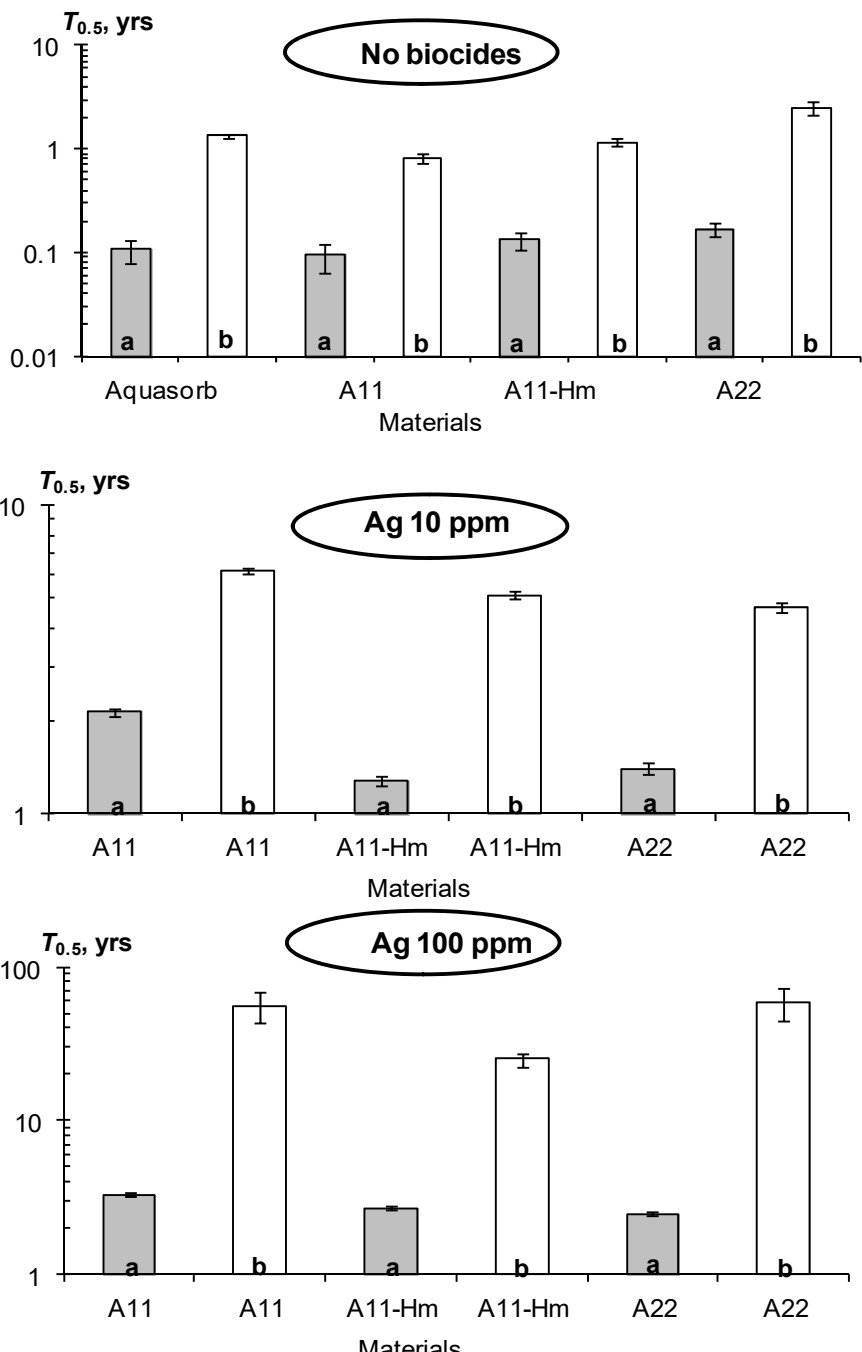

**Figure 3.** Calculated half-life values for composite gel-forming materials without biocides and with ionic silver in a polymer matrix. The liquid phase in the incubation experiment: (a) with compost extract, (b) pure distilled water.

One of the possible ways to reduce the biodegradability of composite polymer materials is to introduce biocidal additives into the polymer matrix. The introduction of silver ions effectively increases the half-life of gel composites in proportion to the biocide dose (Figure 3). A 0.1% dose (10 ppm in swollen gel) increases resistance to biodegradation by 5–10 times, and a higher 1% dose (100 ppm in swollen gel) up to 20–35 times or more. As a result, the half-life of pure gel swelling in distilled water reaches 5–6 years with a 0.1% dose of silver and 30–60 years if using a 1% dose of biocide. Similar values for gel structures in

an aqueous solution containing compost extract varied in the ranges of 1.3–2.1 years and 2.4–3.3 years, respectively.

Gel-forming composite A22 with peat filler had the highest resistance to biodegradation. For distilled water, its half-life was $5.8 \pm 1.9$ years at 10 ppm silver and $63 \pm 24$ years at 100 ppm concentration. The use of compost extract reduced these values to $1.4 \pm 0.2$ and $2.4 \pm 0.7$ years, respectively. The most biodegradable was the A11-Hm composite with a humate filler, possibly due to the partial binding of ionic silver by humates, leading to biocidal deactivation.

## 4. Discussion

### 4.1. Analysis of the BOD Kinetic Approach for Assessing the Biodegradation of Hydrogels

The proposed kinetic approach is based on the balance between the rates of biodegradation of incubated organic polymers and the consumption of oxygen in a system closed to external mass transfer. The oxygen limit in such a system causes a gradual decrease in the rate of biodegradation over time, which is reflected by the basic kinetic Equation (3) in a differential form. Its solution in the form of Equation (4) almost completely coincides with the well-known empirical Larson model [14,26] for describing the mineralization of chemicals. We only put forward a physical explanation of this model in the form of oxygen limitation, which occurs both in closed laboratory incubation systems and in real soil conditions with rather slow diffusion of oxygen from the atmosphere [17]. Further estimation of the kinetic constant of biodegradation and calculation of the half-life of organic material using Equation (8) are identical to those of the Larson approach [26]. The forms of BOD curves obtained in our study (Figures 1 and 2), adequately reproduced by the pseudo first-order kinetic model discussed above (Equation (4), Table 1), are almost identical to BOD curves for plastic degradation in the aquatic environment [16] and carbon dioxide emission curves for biodegradation of composite hydrogels in [27]. These results confirm the universality of the proposed kinetic approach and the possibility of using it for a physically based kinetic assessment of the biodegradability of chemicals based on both BOD analysis and $CO_2$ emission monitoring.

The most problematic aspect, in our opinion, is the assumption of the molar equivalence of carbon in a biodegradable material, consumed oxygen and released carbon dioxide. This assumption was the basis for the transformation of BOD curves into kinetic biodegradation curves (Equation (5)) and standardization of analysis with the calculation of the standard amount of material (Equation (9)). From a chemical point of view, such an assumption may be incorrect, since even for fully aerobic biodegradation (which is not obvious in an aqueous medium and aqueous superabsorbents), the molar ratios of carbon and oxygen may differ depending on the chemical composition of the oxidized organic material. An abstract training example shows that the complete oxidation of one mole of graphite carbon requires one mole of oxygen. However, the mineralization of methane to the final products of $CO_2$ and $H_2O$ requires two moles of oxygen for one mole of methane carbon. For pure acrylic hydrogels, the ratio we use "1 mol C equals 1 mol $O_2$", in principle, is almost maintained. For example, from the chemical formula of polyacrylamide $(C_3H_5NO)n$, it follows that its complete mineralization with the formation of stable gaseous substances ($CO_2$, $H_2O$ and $N_2$) requires 1.25 moles of oxygen for each mole of PAA carbon, that is, a little more than our 1:1 ratio. However, in composite acrylic material-superabsorbents with components in the form of natural biopolymers (peat, humates, polysaccharides, etc.) [7–12], this ratio can certainly shift more towards higher oxygen consumption. However, from a mathematical point of view, this situation should not affect the possibility of transforming BOD curves into biodegradation curves of organic material. Indeed, in Equation (5) proposed for this purpose, only the coefficient 12/32 changes, and the main parameter in the form of the kinetic constant (*b*) remains unchanged. Therefore, the estimation of the half-life of this organic material according to Equation (8) does not change. The only problem here is a possible overestimation of the standard amount of incubated material calculated by Formula (10). In this case, oxygen

deficiency inhibition may occur earlier than expected from the equivalent 1:1 molar ratio, since the amount of potentially biodegradable carbon in the closed system exceeds the amount of available oxygen. This situation can certainly affect the shape of the BOD curve. However, for slowly decomposing materials with quasi-linear BOD curves, especially at the initial stages of the incubation experiment (Figure 2), the available oxygen is quite sufficient, and in this case the BOD analysis clearly reflects the potential biodegradability of these materials, determined by their chemical composition. All the problems discussed above should be considered in the further development of a kinetic approach to assessing the biodegradability of organic chemicals based on BOD analysis.

*4.2. Comparison of the Obtained Experimental Results with Known Data*

The generally accepted point of view emphasizes the high resistance of polymeric acrylic hydrogels to biodegradation. This point of view is reflected in many modern reviews of hydrogels by dividing them into "biodegradable" and "non-biodegradable" [3,7–12]. Moreover, environmentally friendly "biodegradable", predominantly biopolymeric polysaccharide gels or, more rarely, their compositions with synthetic acrylic polymers are usually opposed to purely synthetic polymers alien to the environment [9–11]. We will not discuss here the environmental benefits or risks of synthetic acrylic gels, which have long been effectively used in medicine, agriculture and landscaping, despite the potential carcinogenicity and danger of acrylamide as a raw material for their production [12,28]. There is no doubt that polymers that are absolutely resistant to biodegradation do not exist a priori; therefore, it is necessary to correctly evaluate their life time depending on the chemical composition and environmental conditions of biodegradation.

The known published sources provide conflicting information on the biodegradation of gel-forming polymers, mainly concerning acrylamide and polyacrylamide (PAA), as well as biopolymeric, predominantly polysaccharide hydrogels. For example, Lentz et al. [13] reported a rather slow biodegradation of PAA, not more than 10% per year, and primarily through the shear-induced chain scission and photodegradation. Using the standard exponential biodegradation Model (7) and expression (9) for half-life, it is not difficult to obtain the following formula for estimating half-life based on data on the percentage of biodegradation (%D) for a known period of time ($T_i$):

$$T_{0.5} = T_i \frac{\ln(2)}{\ln[100/(100 - \%D)]}. \tag{14}$$

According to this formula and the data [13], the half-life of PAA exceeds 6.5 years. Closely estimated half-lives of 5–7 years for superabsorbent gel-forming polymers are reviewed by [12]. These estimates significantly exceed our results of $T_{0.5}$ = 1–1.2 years for the radiation-crosslinked technical PAA obtained in our previous publication [2], as well as the results of this research for acrylic composites (see Figure 3). However, some publications report that PAPA and acrylamide degraded in soils very quickly with half-life values of a few days [29–33]. Sojka and Entry [33] reported that PAA was completely degraded within 5 days after applying 0.05% to garden soil. Lande et al. [29] estimated the half-life of acrylamide monomer in agricultural soils as ranging from 18 to 100 h at a concentration of 25–500 mg/kg and a temperature of 20–22 °C. Soil microorganisms are capable of utilizing PAM or acrylamide as a source of nitrogen [30,32,33]. Biodegradation of PAA occurs as microorganisms use the amide group of the polymer as a nitrogen source and/or the carbon backbone as a carbon source [34]. Many microorganisms (*Enterobacter aerogenes*, *Rhodococcus* sp., *Helicobacter pylori*, *Bacillus* sp., *Acinetobacter* sp., *Azomonas* sp., *Pseudomonas* sp., *Chlostridium* sp., etc.) can generate extracellular amidases that successfully utilize the amide groups of polyacrylamide. Aliphatic amidase (cd07565) from *Pseudomonas putida* results in a 46% degradation of PAA after 7 days at 39 °C, as reported in a review [34]. This corresponds to a half-life of PAA no longer than 8 days, according to Formula (14).

The article [35] evaluated the degradation of new biodegradable gel-forming composites for soil conditioning based on natural raw materials (cellulose, clay minerals, and



humic acids) as a function of the synthesis parameters (swelling degree and composition) and environmental conditions, including type of soil and water amount available for the hydration of the hydrogels. The results [28] showed that the working range of the degree of swelling values and the humic acids amount as well as the synthesis parameters had little influence on the stability of the hydrogels compared to the environmental factors, which appear to be the main factors affecting the degradation of these hydrogels. If the untreated control (pure gels) remained stable for 8 days of the incubation experiment (weight loss of 3–5%, statistically significantly no different from zero), then contact with forest soil led to 30–40% losses, and with agricultural soil—to 60–70% losses of polymeric composites due to biodegradation. These losses in terms of half-life offer a range of 0.5 from 6 to 15 days, or significantly less than 1 month. This example independently confirms the most important conclusion of our research: the biodegradability of gel-forming composites depends to a greater extent on soil-biological conditions, and not on the chemical composition of the materials themselves. The main reasons for a significant increase in biodegradation after direct contact with soils and compost or after the addition of aqueous extracts from them, of course, are soil microorganisms and exoenzymes. It is also possible to assume a priming effect, well known in soil biochemistry [36].

This year's review [7] presents data from 22 recent publications on the biodegradability of hydrogels, expressed as % degradation over a certain time interval. For ease of comparison, we have transformed all these data into a universal half-life indicator using Formula (14). Similar values for the European biodegradability standard EN 13432 (90% degradation in 180 days) and the American standard ASTM 6400 (60% in 180 days) [3] were used as reference values, providing a "standard half-life" of 54 and 136 days, respectively. For the entire array of data provided in [7] ($n = 33$), the half-life of polymer hydrogels varied in a wide range from 2 days to 285 years. Using European and American standards, we calculated the probability of a particular hydrogel falling into the "biodegradable" class (half-life is less than 54 or 136 days). Of the entire data set from [7], 61% were conditionally biodegradable according to the American and 33% according to the European classification. If we analyze only composite superabsorbents with acrylic components (acrylates and polyacrylamide in various combinations with each other and natural biopolymers), the probability of their belonging to "biodegradable" is 57% according to the US standard and 24% according to the European standard. For other, mainly biopolymer gel-forming composites based on polysaccharides without the addition of acrylic copolymers, such probabilities of biodegradability are slightly higher (67 and 58%, respectively). Consequently, according to the formal criterion of the USA, more than half of the gel-forming superabsorbents analyzed in [7] are biodegradable, and the presence of acrylic copolymers in their composition has little effect on this indicator, reducing the probability of biodegradability by only 10%. According to a more stringent European criterion, only 24% of gels with acrylic components are biodegradable, compared with 58% of "biodegradable" composites based on natural biopolymers (mainly polysaccharides).

For soil gel-forming conditioners, a half-life reference of 1 year can be introduced, because if 50% of the conditioner degrades in a year, then it is unlikely that its use for increasing water retention or soil erosion protection can be economically viable. An analysis of the review [7] and other publications [2,8,12,27,35] shows that no more than 30% of superabsorbents pass such a test for resistance to biodegradation, and in other cases, the materials lose half their weight (and therefore functionality) within 1 year of use. This serious problem poses a challenge for chemical technologists and geo-engineers to further improve gel-forming composite materials for soil engineering with higher resistance to biodegradability.

### 4.3. Composites with Biocides

The most obvious way to reduce the biodegradation of polymer materials is the introduction of biocidal components into their composition [2,37]. In our previously published works [2,8,22] and in this publication, silver biocides in the form of ions or nanoparticles

were used, apparently for the first time for such a purpose for gel-forming soil conditioners. Known similar developments with silver biocides in the composition of synthetic and biopolymer materials mainly concern medical preparations and antiseptics [37,38]. Silver biocides effectively increase the resistance of hydrogels to biodegradation, which is confirmed by both previously published data [2,8,22] and new results of BOD analysis (see Figures 2 and 3). They are in good agreement with the known data on the effective biocidal action of silver ions and nanoparticles with relatively small (1–100 ppm) doses [39–42]. In our proprietary materials A11Ag and A22Ag, the mass fraction of silver varies in the range of 0.1–1%. When the hydrogel swells to an optimal degree of 1:100, the silver concentration in the gel structure does not exceed 10–100 ppm. Such doses effectively suppress biological activity, including soil and plant pathogens in the rhizosphere [22] and, at the same time, are not dangerous for plants and soil zoocenosis, including earthworms [43]. In the soil, effective concentration of silver suppressing the growth of plants varies in the range of 50–1000 mg/kg of the solid phase, or from 250 to 5000 ppm of the liquid phase at 20% soil water content [2,42]. Shclich et al. [43] reported similar values of silver ions or nanoparticles from 200 to 400 ppm for earthworms in the soil. Even if we assume an unlikely event of leaching of all silver with fast biodegradation of gel-forming composites A11Ag or A22Ag, the concentrations in the soil solution of 10–100 ppm will not exceed the threshold mentioned above, which is dangerous for plants or zoocenosis of the soil.

A higher silver content in composite gel-forming materials (up to 8%, as in antiseptic hydrogel dressings, according to [38]) is obviously not acceptable for soil conditioners due to the risk of exceeding the critical concentration threshold for adverse environmental effects. In review [37], silver is also considered as the most effective biocide for composite materials and coatings, along with zinc, copper, gallium, selenium, halogens (chlorine, iodine), nitric oxide, $TiO_2$ and $TiO_2$-based nanocomposites, etc., as well as antimicrobial peptides/organosilicon surface coatings. Some synthetic polymeric materials and composites for agriculture can exhibit biocidal properties, for example, due to the introduction of phosphonium or sulfonium salts into their structure [9,44]. Due to the relatively high cost of silver, future research is likely to include testing of alternative biocidal additives, for example, based on copper, antimony, phosphonium, ammonium or sulfonium salts, titanium oxide, as well as organic antimicrobial products.

### 4.4. Methodological Aspects of Assessing Biodegradability

Our study was largely related to the methodology of laboratory assessment of the biodegradability of gel-forming materials based on BOD analysis. We have already considered the advantages of this analysis and, first of all, the possibility of simultaneous fully automated computer monitoring of the biodegradation of many samples under different incubation conditions (temperature, water content, chemical composition, etc.). Simple models (11), (12), taking into account the first-order kinetics of biodegradation of materials under standardized conditions, make it easy to calculate the basic indicator of their half-life. In contrast to the indirect prediction of the service life of superabsorbents in the environment based on thermal aging and degradation rate, known in polymer chemistry [45], BOD analysis uses direct incubation experiments with real microbial and enzymatic degradation of the materials under study.

At the same time, BOD analysis has a number of drawbacks. Firstly, it displays a relatively low accuracy of manometric control of oxygen absorption. For a VELP BOD sensor, it is 0.355 kPa or about 1 $g/m^3$ of $O_2$. For comparison, modern IFR-$CO_2$ gas analyzers with direct detection of emitted $CO_2$ due to aerobic biodegradation have an accuracy of 10–50 ppm or, according to Formula (1), a gas concentration of 0.02–0.1 $g/m^3$, that is, a 10–50 times higher accuracy than that of indirect (manometric) BOD analysis. The low accuracy of BOD analysis requires a long incubation time to obtain representative (statistically significantly different from zero) BOD data, up to several months in the case of biodegradation-resistant materials, which is the second main disadvantage of this method.

Another inconvenience is associated with the mandatory use of a $CO_2$ absorber (alkali). The swollen 1:100 hydrogels maintain a water activity close to one or a relative humidity of about 100% in the airspace of a closed vial, which is clearly visible in the photo (Figure 1) in the form of water condensate droplets on the walls inside the glass VELP respirometer. Therefore, the initially solid-phase alkaline absorber (KOH) quickly turns into a liquid, absorbing water vapor from almost saturated air. The absorption of $CO_2$ by a liquid is limited by the diffusion of gas molecules in the volume of the liquid phase, significantly (up to ten thousand times) slower than in the free air space of the pores of the initial solid-phase adsorbent. This situation can weaken the intensity of $CO_2$ uptake in the vial and distort the BOD manometric estimate. In addition, the RESPIROSOFT™ Software, despite the possibility of internal statistical processing of several experiments to calculate average BOD estimates, has a problem with the reproduction of individual BOD curves and their translation into formats convenient for subsequent processing, for example, MS Excel spreadsheets.

All these problems, and primarily the low accuracy and duration of BOD analysis, pose the task of developing alternative methods for instrumental assessment of the biodegradability of gel-forming composites. The most promising, in our opinion, here is the use of IFR analyzers of $CO_2$ dynamics (and in the case of anaerobic destruction also of $CH_4$ dynamics) in similar BOD analysis methods of laboratory incubation. As an example, we refer to the publication of Hiroki et al. [27], who used $CO_2$ respiratory analysis and obtained respiratory curves similar to our BOD analysis (see Figure 2) for a 30-day incubation experiment with composite hydrogels based on carboxymethyl cellulose and starch. However, these experiments used a rather outdated technique of volumetric absorption of the emitted $CO_2$ by soda lime columns with subsequent weighing. We have started a new development based on PASCO equipment (USA, PASCO Scientific, California, Roseville, [46]) that allows fully automated incubation experiments with continuous $CO_2$ control and plan to prepare the next publication on the results of these experiments with composite gel-forming materials for soil conditioning.

## 5. Conclusions

Summarizing the results of the study allows us to formulate the following main conclusions:

- A new methodology for quantifying the biodegradability of organic gel-forming materials based on precision VELP equipment for BOD analysis is proposed.
- This methodology uses the original kinetic model of real-time BOD curves obtained in long-term (60–120 days) incubation experiments with automatic manometric monitoring of BOD dynamics.
- The kinetic constants of the long-term dynamics of BOD and biodegradation of the studied organic materials are identical values underlying the calculation of the half-life of these materials.
- The experiments revealed a strong contrast between half-life values of the studied aquatic superabsorbents depending on their incubation conditions. Pure hydrogels swollen in distilled water have rather high half-life values, ranging from $0.8 \pm 0.2$ to $2.4 \pm 1.6$ years. The addition of a small amount of fresh aqueous extract from organic compost to the liquid phase of the gels sharply (8–15 times) reduces the half-life of these materials up to 33–60 days.
- These results show that acrylic gel-forming superabsorbents of the same chemical composition can, depending on environmental conditions, be either "non-biodegradable" or "biodegradable" according to the European EN 13432 and the American ASTM 6400 biodegradability standards.
- Embedding silver biocides into the polymer matrix of the studied composite superabsorbents effectively (up to 20–35 times) reduces the rate of their biodegradation, which can be used in soil engineering technologies to prolong the service life of these water-saving soil conditioners.

Further quantitative research in this area should be aimed at finding and testing cheaper biocidal components, as well as improving the methodology of laboratory analysis and modeling of biodegradation of polymer superabsorbents for soil conditioning.

**6. Patents**

The results of the work are used in the synthesis technology of biodegradation-resistant filled hydrogels patented in the Russian Federation:

patent RU №2726561 (https://findpatent.ru/patent/272/2726561.html 22 July 2022)
patent RU 2639789 (http://www.findpatent.ru/patent/263/2639789.html 22 July 2022).

**Author Contributions:** A.V.S.: conceptualization, methodology, supervising, formal analysis, software, writing and funding acquisition; N.B.S. and V.I.B.: investigation, data curation, validation, visualization. All authors have read and agreed to the published version of the manuscript.

**Funding:** State Contract of the Ministry of Science and Higher Education of the Russian Federation No. 075-15-2022-1212 "Development and application of innovative soil ameliorants to increase productivity and prevent degradation of arid lands".

**Conflicts of Interest:** The authors declare no conflict of interest. The funders had no role in the design of the study; in the collection, analyses, and interpretation of data; in the writing of the manuscript, as well as in the decision to publish the results.

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
