# Peer review of "Biodegradation of Aqueous Superabsorbents: Kinetic Assessment Using Biological Oxygen Demand Analysis"

_jcs, doi:10.3390/jcs7040164_

Round 1
Reviewer 1 Report
The detailed review comments are attached at Review's comments___v2.

Author Response
Dear Reviewer #1! We sincerely thank you for your work on reviewing the manuscript, its positive assessment and valuable comments on its improvement. We have tried to change the manuscript, according to your comments, where possible. For other comments and suggestions, we give the following explanations in the order of numbering in your review.
- We agree, the conclusion is removed, the "Introduction" is modified.
- We cannot accept this proposal, since the development of a theory and kinetic model for BOD analysis is one of the research tasks. Accordingly, the results were obtained for it and they are placed in the "Results" section. To resolve this issue, we have highlighted the purpose and objectives of the study in the "Introduction".
- We agree, the BOD abbreviation stands for both in the abstract and in the text (line 62).
- We, of course, agree, but could not find the reasons for your remark. All formulas were written the same way in the Word formula editor.
- We agree and added the assignment of models (4) and (12): "for BOD dynamics".
- We tried to partially expand the explanations in the captions (Fig. 1, 2), although this remark remained unclear to us.
- The remark about the Table also remained unclear to us. After all, the text explained the meaning of Table 1 and the subsequent analysis of its contents (lines 347-358 in the new version of the manuscript).
- This is an understandable question. We used all the information available to us and, believe me, very few published information to compare the results obtained with those known in international science. This is a mandatory requirement for any scientific publication. Regarding the comment on the analysis of the model, we took it into account in the new version of the manuscript, adding the relevant information to the "Discussion" Section (lines 409-462).
9 Thank you, we corrected all the grammatical errors and typos you noticed and tried to spell the manuscript again. Not being native speakers, we also hope for the help of the proofreaders of the article, which is traditional for MDPI.
March 20, 2023
Once again with gratitude and respect from the team of authors, prof. A.V. Smagin

Reviewer 2 Report
In this work, the authors investigate the biodegradation of gel-forming composites for soil engineering: laboratory BOD analysis. The work is very interesting and with a high impact on literature. The manuscript can be accepted for the publication in J Comp Sci after addressing the following major comments.
1. The title i not very suggestive. Please reformulate.
2. Abstract: Describe more relevant results in the abstract. Mention the purpose for which this study was conducted.
3. The lengthy sentences may be split in to smaller sentence without change of its meaning.
4. Also, suggested to include the recent references in the introduction part.
5. Introduction: There are works that deal with the topic of organic matter that should be cited in the introductory part of the first paragraph. Update the literature with the latest articles in the field: Environmental Pollution 267, 2020, 115409, Environmental research 182, 2020, 109136, Water Environment Research 90(3):(2018) 220-233.
6. The results and discussions part should be compared with the literature data. To redo the part of results and discussions by a systematic presentation of the results by which the readers of the articles manage to follow the article more easily..
7. Figure quality is poor throughout. To improve the quality of the figures. Enlarge the characters in the figures.
8. To correlate the results obtained with the results present in other works.
9.Conclusions should be short with important observations.
10. It is a complex study but I have some objections to the structuring of the manuscript. At the end of the introduction, the authors described the purpose (objectives) of the study, mentioning also the methods used, obviously briefly, but this should be explained in more detail in the methods. In terms of materials and methods, each technique applied should appear as a subpoint of this chapter, so it would be easier to follow the results as well. In the conclusions appear aspects that should appear in the chapter of results and be described only strictly the conclusions of this study. Do you consider that the topic is relevant in the research area and if so, what is it? In case of conclusions, you could explain what does it add to the subject area compared to other published material?
Author Response
Dear Reviewer #2! Thank you very much for your positive feedback and valuable comments aimed at improving the manuscript. We have tried to take into account some of them in the new version of the manuscript, and for the controversial, in our opinion, comments, we give explanations of our position below, following the numbering according to your review..
- We agree and tried to reformulate the title.
- We have also rewritten the abstract, specifying the purpose of the study and its relevant results, according to your comment.
- Where possible, we revised the sentences, reducing their length without violating the meaning.
4 and 5. We have rewritten the "Introduction" to include more relevant up-to-date information and highlighting the purpose and main objectives of the study. Unfortunately, we could not use your references (Environmental Pollution 267, 2020, 115409, Environmental research 182, 2020, 109136, Water Environment Research 90(3):(2018) 220-233), since we found there only the works of highly respected Romanian colleagues on the pollution of aquatic ecosystems, too far from the topic of our research (biodegradation of synthetic hydrogels, theory and practice of BOD analysis). You probably accidentally made a typo by highlighting these publications highlighting these works, perhaps from another of your reviews. Or we didn't understand what you wanted to advise us to take from there for our "Introduction".
- We believe that we have compared our results in sufficient detail with the literature data known in this field. Nevertheless, we have added another paragraph (4.1) to the "Discussion" in order to strengthen the analysis of the theoretical part of the work.
- The quality of the figures in the manuscript fully corresponds to the requirements of J. Campos. Sci. You can see this by enlarging our figures and photos. For example, the photo on Figure 1 when zooming in (and this is a standard option in electronic documents) even allows you to see the condensation droplets inside the glass BOD-respirometer. Nevertheless, we took into account your remark by enlarging the symbols on Figure 2.
- This remark, essentially duplicating remark 6 already made earlier, remained incomprehensible to us. We, let's say it again, performed an extensive comparison of the obtained results with the data known to us published by the international scientific community in this field (see the updated Discussion Section and the References list of 46 publications)..
9 We took into account this comment, concretizing the conclusions in accordance with the objectives and results of the study and highlighting them.
- We also tried to take into account the general comments of paragraph 10 in the new version of the manuscript, along with the comments of Reviewer No. 1. The relevance of the research is set out in the Introduction, where the purpose, main tasks and novelty of the research are also highlighted separately. The section "Materials and Methods" provides general information about the studied synthetic superabsorbents, BOD analysis and precision VELP equipment. The specific part concerning the theory of BOD analysis and its kinetic modeling has been specially placed by us in a separate paragraph of the Results Section, since these are methodological RESULTS obtained in the work in accordance with its tasks. In the formulation of novelty (Introduction), in the Discussion and Conclusions, we highlight what this work has given us, namely, a new methodology for BOD analysis of the biodegradability of gel-forming organic materials, as well as the conclusion that their biodegradability strongly depends on the experimental conditions, in particular, the addition of compost extract with decomposing microorganisms for organic matter.
March 20, 2023
We hope that the new version of the manuscript satisfies most of your comments. Once again, with gratitude for your work on the revision of the manuscript and with best wishes from the team of authors, prof. A.V. Smagin.

Reviewer 3 Report
Authors have conducted worthwhile piece of work. The present investigation possess substantial significance towards the mitigation of pollutants.
The authors have investigated the role of super-adsorbents in the biodegradation study. The study is well-designed, planned, and executed appropriately. The manuscript is well written so that the readers and researchers get benefitted and their vision will be augmented. The purpose of the biodegradation study is clearly narrated by the authors. Authors have utilized one significant influencing parameter i.e. BOD. The study has been conducted in full length from 40-60 days so comprehensive information can be obtained for further standardization of the analysis. As a matrix authors have employed a gel-forming matrix which is an important attribute of the study. Moreover, they employed kinetic models to understand the mechanism mathematically. I would like to appreciate their approach to utilizing models in their study. Additionally, the research content has been patented and it has been mentioned by the authors as well in the manuscript. The Discussion and Conclusion are nicely written.
Round 2
Reviewer 1 Report
Accept for publication.
Reviewer 2 Report
- The manuscript is not presenting anything novel.
- It is more like a technical note.
- The language of the whole manuscript is very good and an improvement on the English language needs to be done in order to improve language of the manuscript.
- The structure of the manuscript need to reconstruct.
- The abstract need to rewrite and the author/s should focus on the main findings and the main results.
- The results need more explain and discussion where the current discussion dose not enough and the author/s only describe the results without any interpretation.
- Additional figure and should be add.
- The author/s should be try to compare the results of the current study with the results of other researchers and it is required to explain if there is any agreement or disagreement.